# Effect of Different Sealers on the Cytocompatibility and Osteogenic Potential of Human Periodontal Ligament Stem Cells: An In Vitro Study

**DOI:** 10.3390/jcm12062344

**Published:** 2023-03-17

**Authors:** Shehabeldin Saber, Shereen Raafat, Mohamed Elashiry, Ahmed El-Banna, Edgar Schäfer

**Affiliations:** 1Department of Endodontics, Faculty of Dentistry, The British University in Egypt (BUE), El Sherouk City 11837, Egypt; 2Centre for Innovative Dental Sciences (CIDS), Faculty of Dentistry, The British University in Egypt (BUE), El Sherouk City 11837, Egypt; shereen.nader@bue.edu.eg; 3Department of Endodontics, Faculty of Dentistry, Ain Shams University, Cairo 11566, Egypt; dr.ashmmk@gmail.com; 4Pharmacology Department, Faculty of Dentistry, The British University in Egypt (BUE), El Sherouk City 11837, Egypt; 5Department of Biomaterials, Faculty of Dentistry, Ain Shams University, Cairo 11566, Egypt; doctorbanna@asfd.asu.edu.eg; 6Central Interdisciplinary Ambulance in the School of Dentistry, University of Münster, 48149 Münster, Germany; eschaef@uni-muenster.de

**Keywords:** ADSeal, calcium silicate-based sealers, cytocompatibility, Endosequence HiFlow, GuttaFlow-2, osteogenic differentiation, periodontal ligament stem cells, VDW.1Seal

## Abstract

Background: There is tendency for unavoidable sealer extrusion in some clinical cases. This might adversely affect host stem cells and affect healing. This study aimed to investigate the effect of different sealers on the cytocompatibility and osteogenic potential of human periodontal ligament stem cells (*h*PDLSCs). Methods: The cytotoxic effect of the extracted elutes of VDW.1Seal (VDW.1), Endosequence BC Sealer HiFlow (ES), GuttaFlow-2 (GF), and ADSeal (AD-S) on the *h*PDLSCs was determined using the MTT assay. Cell proliferation and migration were assessed by the scratch wound healing assay. Osteogenic differentiation potential was assessed. Measurement of pH values and calcium ions release was performed. Results: GF had a significantly higher percentage of viable cells. The cell migration assay showed that GF demonstrated the lowest open wound area percentage. GF and AD-S showed the highest calcium nodule deposition. GF demonstrated higher ALP activity than ES. Expression of RUNX2 and OC genes was similar for all sealers, while OPG gene expression was significantly higher for VDW.1 and GF. ES and AD-S displayed the highest pH values on day 1. Calcium ion release of ES and VDW.1 was significantly the highest. Conclusions: GuttaFlow-2 and VDW.1Seal sealers have favorable behavior toward host stem cells.

## 1. Introduction

Root canal treatment involves several procedures that should be performed thoroughly and with utmost care to ensure the best outcome [1]. Root canal obturation is the final procedure during treatment, where the root canal system becomes filled three-dimensionally using core materials and sealers [2]. Unfortunately, geometric dissimilarity exists between root canals and the filling points. To address this, obturation techniques were modified to use auxiliary cones besides the master cone, as with the cold lateral compaction technique, or heat and pressure, as with the warm vertical compaction technique described by Schilder (3), who suggested the cementodentinal junction as the point where the filling should be stopped as filling to this point fills the canal without impingement on the periapical tissues and enhances the physiologic closure of the root canal with cementum [3]. Another approach is to use a carrier coated with gutta percha as a single cone. It is well known that the hydrodynamic nature of some of these techniques might result in the accidental extrusion of obturation materials beyond the apical foramen. [4]. Although the contact between these materials and the periapical tissues may not necessarily produce an unfavorable tissue reaction as many host factors are involved in this process, it might yet compromise the process of healing or elicit a local inflammatory response that may jeopardize the success of root canal therapy [5,6]. 

The biological characteristics of endodontic sealants were evaluated in vitro using animal cell lines such as RPC-C2A rat pulp cells, L-929 mouse fibroblasts, osteoblastic cells, and more recently, human periodontal ligament stem cells (*h*PDLSCs) [7]. *h*PDLSCs are a multipotent stem cell population that exhibits typical properties of bone-marrow-derived mesenchymal stem cells (BMMSCs) and have the ability to differentiate into cementum/PDL-like-tissue-producing cells in vivo [8].

Human periodontal ligament stem cells could renew themselves and display cell surface markers that are comparable to those seen on BMMSCs. These markers include CD105, CD90, CD73, CD59, CD29, CD13, and CD10, and they do not express CD45, CD34, CD14, or HLADR. [8,9]. The *h*PDLSCs are not only important because of their differentiation potential, but they also have an immunosuppressive effect [10]. It was found that these stem cells can inhibit the proliferation of B and T lymphocytes [11] and shift macrophages polarization toward the M2 phenotype [12], thus causing tissue homeostasis and direct the inflammatory reaction to proceed towards the healing phase [13]. Accordingly, the extrusion of a root canal sealer should not adversely affect these cells.

Hydraulic calcium silicate-based (HCSB) materials are recognized for their superior sealing ability, biocompatibility, and bioactivity [14]. They have been recommended for use clinically in vital pulp therapy, root repair, root-end filling, regenerative endodontic procedures, or sealers [15,16]. The potential of these materials to be used as sealers led to an upsurge in the manufacturing of new products with different composition modifiers to improve their physical and biological properties [17]. 

As the initial phase in determining whether or not newly developed endodontic sealers are safe, cytotoxic studies in vitro are being conducted on these materials [18]. This study intended to assess the effect of the new HCSB sealers VDW.1Seal (VDW, Dentsply Sirona, Munich, Germany) and Endosequence BC Sealer HiFlow (Brasseler USA Savannah, GA, USA) in comparison with a polydimethylsiloxane-based sealer (GuttaFlow-2, Coltene/Whaledent, Langenau, Switzerland) and a resin sealer (ADSeal, Meto Biomed, Cheongju, Korea) (Table 1) regarding their cytocompatibility and osteogenic differentiation potential of *h*PDLSCs. 

## 2. Materials and Methods

This in vitro study manuscript complies with the “Preferred Reporting Items for Laboratory studies in Endodontology (PRILE) 2021” criteria [19].

### 2.1. Materials Preparation

The detailed composition of the materials used is listed in Table 1.

Materials were prepared as discs (5 mm diameter and 2 mm high) according to the manufacturer’s instructions under aseptic conditions, left undisturbed to set at 37 °C and 5% CO_2_ and 95% relative humidity for 48 h, and sterilized using ultraviolet light for 30 min on every side. The eluates of the different materials were extracted and prepared according to the International Organization for Standardization (ISO). Serial dilutions at 1:2 and 1:4 ratios of the extraction medium were prepared, and the material surface area/medium volume ratio were made at 3 mm^2^/mL, approximately, per ISO10993-12 standards [20]. 

### 2.2. Periodontal Ligament Stem Cells Isolation

Permanent human molars were obtained from healthy donors for orthodontic reasons at the Oral Surgery Department, Faculty of Dentistry, after the approval of the Ethical Committee (FD BUE REC 22-005, approval date from 23 January 2022 to 23 January 2023). Informed consent was obtained from the patients after a clear explanation by the principal investigator regarding the purpose of the study and that it is a voluntary action serving the field of research instead of wasting the extracted tissue by it being incinerated. The principal investigator confirmed the patient’s data confidentiality. 

The extracted teeth (n = 10) were devoid of any carious lesions and the patients selected did not suffer from serious inflammation around the extracted teeth. After extraction, the teeth were stored, and stem cell isolation was performed as described by Dahak et al. 2020 [21]. Briefly, periodontal ligaments were cut into tiny pieces and digested for 1 h at 37 °C using a mixture of 3 mg/mL collagenase and 4 mg/mL dispase II in the culture medium, which consisted of DMEM/F12 (Dulbecco’s Modified Eagle Medium/F12 Ham medium, Sigma) supplemented by 10% FBS (fetal bovine serum, Gibco BRL, CA, USA) and 1% antibiotic/antimycotic (Gibco). Cells were then incubated in 5% CO_2_ at 37 °C in humidified atmosphere. Observation of cell development and morphology was performed using an inverted microscope (TCM 400, Labomed, Los Angeles, CA, USA). Cells at the fourth passage were used in this study and at least three triplicates for each group were performed.

### 2.3. Characterization and Identification of Isolated PDL Cells 

#### 2.3.1. Characterization by MSC Cell Surface Markers Expression

The *h*PDLSCs were detached and identified by immunophenotyping with primary antibodies raised against CD45, CD34, CD73, CD90, CD105, and HLA-DR for 1 h. Immunophenotyping procedures were performed as previously described by Seo et al., 2004, and Feng et al., 2010 [9,22]. 

#### 2.3.2. Multilineage Differentiation Potential

The multilineage differentiation ability of *h*PDLSCs was analyzed in vitro in a 24-well plate using the commercially available kit (Identification kit, R&D Systems, Minneapolis, MN, USA). Wells were analyzed using an inverted microscope (TCM 400) and imaging was performed using a digital camera (Canon, Woodhatch, UK).

### 2.4. Cell Viability Assay

The cytotoxic effect of the extracts was determined by the 3-(4,5-dimethylthiazol-2-yl)-2,5-diphenyltetrazolium bromide (MTT) assay. Briefly, as stated in previous experiments, cells were cultured and incubated in Dulbecco’s modified eagle medium (DMEM; Sigma, St. Louis, MO, USA) (n = 4 per sealer). The next day, sealer extracts were added, and cells were incubated for 24, 48, or 72 h. At the determined time periods, 1 mg/mL of MTT was added for 4 h after medium removal. Then, dimethyl sulfoxide (DMSO) was added to dissolve the violet deposits produced as a result of MTT reduction by the viable cells. The values of the optical density (OD) were measured at 570 nm [20]. Cell viability was determined using the formula by Dahake et al., 2020. [21].

### 2.5. Scratch Wound Assay

Cell proliferation and migration were assessed by the scratch wound healing assay [23]. Briefly, *h*PDLSCs were seeded in sterile 6-well plates (n = 3 per dilution) and left to proliferate until 95–100% confluence. Cells in the medium only were employed as the negative control. A wound was made in each using a sterile P-200 pipette tip, and then eluates of the different materials were added over the attached *h*PDLSCs for 3 days. The scratches of all groups were imaged at 0, 24, 48, and 72 h with a phase-contrast inverted microscope. The area of the wound was determined by computer software (Image J, National Institutes of Health, Bethesda, MD, USA), and the wound closure percentage was calculated for each well. Each group was evaluated in triplicate and the experiment was repeated three times.

### 2.6. Induction of hPDLSCs Osteogenic Differentiation

For osteogenic differentiation, cells were cultured in a 6-well plate with different undiluted extract materials. On the next day, the medium was replaced with a differentiation medium (OsteoDiff medium; Miltenyi Biotec, San Diego, CA, USA) composed of MEM-Alpha, 10% FBS, ascorbic acid-2-phosphate 2.5 mg/L (Sigma Aldrich, Steinheim, Germany), dexamethasone 0.1 μM (Sigma), and beta-glycerophosphate 10 mM (Merck, Darmstadt, Germany). The differentiation was performed for 14 days. The negative control group consisted of cells cultured in a normal medium, and the positive control group consisted of cells cultured in the OsteoDiff medium. Both the osteogenic induction medium and normal medium were changed twice per week.

### 2.7. Cell Mineralization Assay 

Mineralization assay was carried out after 14 days of induction as described by Dahake et al., 2020 [21]. The cells were fixed using 70% (*v*/*v*) ethanol and stained with 40 mM Alizarin Red S (Sigma) for 30 min. Excess stain was removed and cells were observed by an inverted microscope. The staining was solubilized with 10% glacial acetic acid solution producing yellow color, and the absorbance was measured at 405 nm. As previously mentioned, the negative control group consisted of cells cultured in a normal medium, and the positive control group consisted of cells cultured in OsteoDiff [24].

### 2.8. Assessment of Alkaline Phosphatase (ALP) Enzyme Activity

ALP activity of *h*PDLSCs of each specimen was assessed after differentiation for 14 days using a commercially available kit (LabAssayTM kit, Wako Pure Chemicals, Tokyo, Japan) according to the manufacturer’s procedures. Wells were washed to remove unattached cells, after which 0.1% Triton in 1 mL PBS was added. The cell layers were detached and then sonicated for 30 s. Para-Nitrophenylphosphate (p-NPP) (Sigma Aldrich, Steinheim, Germany) was added to the supernatant for 90 min after the removal of cell debris by centrifugation. ALP transforms p-NPP to yellow para-Nitrophenol (p-NP), which was measured photometrically at 405 nm [25].

### 2.9. Osteogenic Gene Marker Expression 

A series of osteogenic markers expressed as runt-related transcription factor 2 (RUNX2), osteocalcin (OC), and osteoprotegerin (OPG) was determined by RT-qPCR performed as a measurement of cell differentiation. The total RNA-containing mRNAs were isolated from lysed cell samples using Trizol (Zymo Research Corp. Irvine, CA, USA), performed according to the manufacturer’s instructions, and cDNA synthesis was performed using a miScript Reverse Transcription kit (QIAGEN), and a miScript PCR kit was used for mRNA quantification. The expression of mRNA was normalized to the expression of β-actin. A list of the primers used in this experiment is supplied in Table 2.

### 2.10. Hydroxyl and Calcium Ion Leaching Evaluation

A 3 mm diameter polyethylene sterile catheter was sectioned horizontally at 2 mm intervals to give a circular mold used for sealer disc preparation. Each mold was ligated with a non-waxed dental floss. Each sealer was injected inside the polyethylene mold, and the resulting disc-shaped specimens measured 3 mm in diameter and 2 mm in thickness (n = 5). Then, each disc was immersed individually in a sealed sterile glass test tube containing 5 mL deionized water (Water HPLC grade, Chemlab, Belgium) and incubated at 37 °C. The leachates were collected from each tube on the 1st, 7th, and 14th days for Ca^2+^ release and pH measurements. Afterwards, samples were moved to clean tubes filled with sterile deionized water.

The pH measurement was carried out using a pH meter. Once the specimen was removed and the storage tube was shaken for 5 s, 1 mL of each leachate was aspirated and placed in the pH meter’s lens. The pH of each sample was determined twice. Inductively coupled plasma/optical emission spectrometry was used to assess Ca^2+^ release from the tested materials at predefined time intervals (n = 5 per sealer) (ICP-ms AGILENT 8800, USA) [26,27].

### 2.11. Statistical Analysis

In the current study, a one-way ANOVA test was performed and Tukey’s post hoc test by using Graph-Pad Prism v8.1.0 (GraphPad Software, San Diego, CA, USA). Data were expressed as means and standard deviations (SD). Values at *p* < 0.05 were considered significant.

## 3. Results

### 3.1. Isolation and Characterization of Periodontal Ligament Stem Cells

Immunophenotyping results showed that cells >95% of *h*PDLSCs possessed surface markers specific for mesenchymal stromal cells, i.e., CD105, CD90, and CD73. The cells possessed very low expression (<5%) of the hematopoietic origin markers such as CD34, CD 45, and HLA-DR (Figure 1a).

Additionally, *h*PDLSCs demonstrated a multilineage differentiation potential into adipogenic, chondrogenic, and osteogenic precursors. Special staining demonstrated the formation of lipid droplets by Oil Red O, mucopolysaccharide by Alcian Blue, and calcium-rich deposits by Alizarin Red (Figure 1b–d).

### 3.2. Cell Viability Assay

There were significant differences as indicated by the statistical analysis in the undiluted elutes for day 1 (*p* = 0.0028), day 3 (*p* = 0.0044), and day 7 (*p* < 0.0001). At day 1, only GuttaFlow-2 showed a significantly higher mean percentage of cell viability in comparison to VDW.1Seal (*p* < 0.05); however, no significant differences were found between the other sealers (*p* > 0.05). At day 3, GuttaFlow-2 showed significantly higher cell viability than VDW.1Seal and ADSeal (*p* < 0.05), with no significant difference compared (*p* > 0.05) to Endosequence BC Sealer HiFlow. Endosequence BC Sealer HiFlow revealed significantly higher cell viability than ADSeal (*p* < 0.05) with no significant difference compared to VDW.1Seal (*p* > 0.05). On day 7, GuttaFlow-2 showed the highest mean percentage of cell viability, while Endosequence BC Sealer HiFlow showed the least percentage of cell viability (*p* < 0.05). No significant difference was found between day 1, day 3, and day 7 for all sealers (*p* > 0.05) except for Endosequence BC Sealer HiFlow, for which the day 3 percentage of cell viability was significantly higher than that for day 7 (*p* < 0.05) (Figure 2a).

For the dilution 1:2, no significant differences were found on day 1 (*p* = 0.0810), on day 3 (*p* = 0.3309), and on day 7 (*p* = 0.5246). The differences between the different time periods examined for all sealers were insignificant (*p* > 0.05) (Figure 2b). For the dilution 1:4, significant differences were found between day 1 and day 7 (*p* = 0.0049, *p* = 0.0061, respectively). GuttaFlow-2 showed the significantly highest percentage of cell viability (*p* < 0.05), while the significantly lowest cell viability was found for ADSeal (*p* > 0.05). No significant difference was found between the sealers on day 3 (*p* = 0.1157). The differences between the different time periods examined for all sealers were insignificant except for Endosequence BC Sealer HiFlow, where the same trend as the undiluted elutes was found (Figure 2c). 

### 3.3. Cell Migration Assay 

In the first 24 h, stem cells exposed to undiluted elutes of GuttaFlow-2 showed the significantly lowest percentage of open wounds compared to all other sealers, indicating a higher migration of stem cells into the scratched area (*p* = 0.0347). All sealers showed a similar performance after the first day of the experiment (48 h *p* = 0.2027, 72 h *p* = 0.2548). For the dilutions 1:2 (24 h *p* = 0.5828, 48 h *p* = 0.0969, 72 h *p* = 0.1001) and 1:4 (24 h *p* = 0.9129, 48 h *p* = 0.1479, 72 h *p* = 0.992), no significant differences were found at any observation time. In general, the percentage of open wounds decreased for all sealers over time (Figure 3 and Figure 4).

### 3.4. Mineralization Assay

The results of calcium nodules deposition, as indicated by the absorbance Of Alizarin Red stain, revealed that GuttaFlow-2 and ADSeal showed significantly higher absorbance than the other sealers, including the negative control group (*p* < 0.05), and were comparable to the positive control group (Osteo). Endosequence BC Sealer HiFlow showed the significantly lowest absorbance (Figure 5a,b; *p* < 0.0001).

### 3.5. Alkaline Phosphatase Activity Assay

The alkaline phosphatase activity was significantly higher for GuttaFlow-2 than for Endosequence BC Sealer HiFlow (*p* < 0.05). The results for all the sealers were comparable to the Osteo group (*p* > 0.05) and significantly higher than the control group (Figure 6; *p* < 0.0001).

### 3.6. Osteogenic Marker Gene Expression

All sealers examined showed similar relative RUNX2 gene expression (*p* > 0.05), which was higher than the control group but lower than the Osteo group (Figure 7a; *p* < 0.0001). Regarding the OPG gene expression, VDW.1Seal showed significantly higher gene expression than Endosequence BC Sealer HiFlow and ADSeal (*p* < 0.05) but a similar expression compared to GuttaFlow-2 (*p* > 0.05). However, for all sealers, the expressions were lower than in the Osteo group (*p* < 0.0001). Regarding the OC gene expression, the same trend as for the RUNX2 gene expression was found; the expressions for all sealers were comparable (*p* > 0.05) and significantly lower than in the positive control group (Figure 7b,c; *p* < 0.0001).

### 3.7. pH-Values

On the first day, pH values for Endosequence BC Sealer HiFlow and ADSeal were significantly higher than for the other sealers (*p* < 0.001). After the first week, pH-values of all sealers were comparable (*p* = 0.8116), while after 14 days, Endosequence BC Sealer HiFlow showed the significantly highest and GuttaFlow-2 showed the lowest (*p* < 0.0001) pH-values (Figure 8).

### 3.8. Calcium Ion RELEASE

Endosequence BC Sealer HiFlow showed the significantly highest calcium ion release on day 1 *(p* = 0.0023), at week 1 (*p* = 0.0017), and at day 14 (*p* < 0.0001). VDW.1Seal showed higher calcium ion release than the other two sealers at day 1 and week 1 (*p* < 0.05). However, on day 14, ADSeal displayed comparable results to VDW.1Seal (*p* > 0.05). GuttaFlow-2 showed the significantly lowest calcium ion release during the entire observation period (Figure 9).

## 4. Discussion

This study aimed to assess in vitro the effect of the newly introduced HCSBS VDW.1Seal, which is identically composed as the AH Plus bioceramic sealer (Dentsply Sirona, Konstanz, Germany) (Maruchi (Anaheim, CA, USA). Material Safety Data Sheet—White ENDOSEAL MTA. 28 May 2021), in comparison to Endosequence Hi-Flow, GuttaFlow-2, and ADSeal regarding the viability and osteogenic differentiation potential of *h*PDLSCs. Investigation of the impact of such materials on *h*PDLSCs, in particular, is important because this cell type is likely to come in contact with a sealer in case of sealer extrusion during obturation with any of the techniques that were stated previously [28]. Additionally, the interaction between the sealer and *h*PDLSCs depend on specific host factors and so may not always exert an adverse effect; however, this interaction in some cases may affect the biological activities of the stem cells thereafter, distressing the healing process and the overall outcome. 

The advantages of such in vitro biologic assays include a repeatable, simple research technique, the prospect of concurrent assessment of several materials under similar conditions, the use of small quantities of tested materials, and a shorter period of testing in comparison to in vivo experiments. In order to ensure the best possible reproducibility and homogeneity between similar in vitro studies, the present investigation followed the specifications summarized in the “Preferred Reporting Items for Laboratory studies in Endodontology (PRILE) 2021” guidelines [19]. The limitations of in vitro studies, however, include oversimplification of methodology and do not fully reflect the performance of these materials clinically. Several variables, such as changes in oxygen levels and local pH or the patient’s immune status, could influence the results observed under clinical or laboratory conditions [29,30,31,32].

ADSeal was chosen as a reference group instead of AH Plus (Dentsply Sirona, Konstanz, Germany), the gold standard of epoxy resin sealers, because previous studies showed that this sealer displayed better biocompatibility than AH Plus [33,34]. Although both are epoxy resin sealers, the superior biocompatibility of ADSeal is linked to the presence of calcium phosphate, which induces bone formation [35]. Additionally, the polymerization reaction of AH Plus is associated with a minor release of formaldehyde [36], which might explain why AH-Plus has been shown to be cytotoxic when freshly mixed but not when set [37,38]. Additionally, another study reported that ADSeal was more biocompatible than AH Plus in an MTT assay on L929 mouse fibroblasts, regardless of being set or unset [39].

The cytocompatibility of the sealer eluates to the *h*PDLSCs was assessed using the MTT assay as a quantitative measure of cell viability and the cell migration assay as a measure of cell proliferation and migration. The MTT-assay has been extensively used in previous similar studies [37,38].

All tested materials in the present study were incubated as set eluates. This agrees with most in vitro studies on the biological interaction between HCSBS and dental stem cells [40]. Although the use of freshly mixed eluates can well foretell the biological response of stem cells, the use of set material eluates is superior to foresee their long-term or delayed response [41]. Future studies should include both preparations in order to provide a comprehensive biological description of the examined materials [42].

The osteogenic differentiation potential of *h*PDLSCs was assessed by three assays: the cell mineralization assay through staining of the calcium-rich extracellular matrix produced by the newly differentiated osteoblasts by Alizarin Red S staining [43]; the alkaline phosphatase activity, which is proportional with the rate of differentiation of *h*PDLSCs toward osteoblasts [44]; as well as by monitoring the expression of target osteogenic gene markers (RUNX2, osteocalcin, and osteoprotegerin). The master regulator of osteoblast differentiation is RUNX2, and its expression is fundamental for mesenchymal stem cells’ differentiation to the osteoblast lineage [45]. The second marker is osteocalcin, which encodes a highly abundant bone protein secreted by osteoblasts to modulate energy metabolism and regulate bone remodeling. The third gene marker is osteoprotegrin, which has a role in the RANK/RANKL/OPG axis and has been shown to be critical in the inhibition of osteoclast production and bone resorption [46]. 

According to the ISO 10993-5 standards, sealers are considered cytotoxic if cell viability is below 70%. The results of the MTT assay demonstrated that *h*PDLSCs cultured with GuttaFlow-2 had the highest percentage of cell viability compared with the other sealers at all observation points and regardless of the medium dilution. This observation is in accordance with previous results showing that GuttaFlow-2 was less cytotoxic on L929 murine fibroblasts compared to MTA Fillapex (Angelus, Londrina, Brazil) and AH Plus [47] and that GuttaFlow-2 displayed less cytotoxicity on human gingival fibroblasts in comparison to ProRoot MTA (Dentsply Sirona), RealSeal sealer (SybronEndo, Orange, CA, USA), and AH Plus [48]. Moreover, GuttaFlow-2 showed no cytotoxic effects on 3T3 fibroblast cells [49]. 

GuttaFlow-2 is a cold, silicon-based flowable system for canal filling that consists of gutta-percha powder with a particle size of less than 30 μm, polydimethylsiloxane, a platinum catalyst, zirconium dioxide, and micro-silver as a preservative and to add an antibacterial effect [47]. The favorable biologic response displayed by GuttaFlow-2 can be attributed to the presence of micro-silver added to replace the relatively cytotoxic nano-silver that was present in the older formulation of GuttaFlow (Coltene Whaledent) [50]. The superior cytocompatibility of the micro-silver is due to the presence of a lesser number of atoms on the surface of these microparticles, causing less reactivity with the surrounding environment [51]. Moreover, GuttaFlow-2 supported the survival and attachment of PDL fibroblasts, which increased over time [50].

The Endosequence BC Sealer HiFlow is a modified formulation of Endosequence BC Sealer (Brasseler USA, Savannah, GA, USA) to be used with warm obturation techniques. Even at higher temperatures, Endosequence BC Sealer HiFlow showed excellent flow, reduced viscosity, and the functional groups of this sealer were not altered [52]. Based on the findings of the previous investigation, Endosequence BC Sealer HiFlow showed good biological properties in terms of cell viability [35], cell migration, adhesion, promotion of mineralization, and ALP activity as well as upregulation of RUNX2 expression by *h*PDLSCs [53,54].

A recent study compared the biocompatibility of AH Plus bioceramic, which is identically composed as VDW.1Seal investigated in the present study, and AH Plus and found that AH Plus bioceramic showed better cytocompatibility and higher osteo/odonto/cementogenic potential than the epoxy resin-based AH Plus sealer [31]. However, in the present study, no such differences were found between VDW.1Seal and the epoxy resin-based sealer ADSeal except for the dilution of 1:2, where VDW.1Seal groups showed a higher percentage of cell viability. This discrepancy may be explained by the better biological effects of ADSeal compared to AH Plus, as pointed out above, and the difference in the setting mechanisms between these two epoxy resin-based sealers, as the sealer setting mechanism, may play an important role in the effect of sealers on stem cells viability [55]. 

The deposition of calcium nodules, an indicator of osteogenic differentiation, was investigated using Alizarin Red S staining [40]. GuttaFlow-2 and ADSeal showed significantly higher calcified nodule formation than the control group and the calcium silicate-based sealers and similar results compared to the Osteo group (Figure 5a,b). These findings contradict the results of different previous studies that reported an increased calcium nodule formation for calcium-silicate-based sealers [40,56].

An increase in the expression of the RUNX2 gene is an indicator of higher osteogenic potential [45]. In this study, although all sealers showed lower gene expression than the Osteo group, expressions were significantly higher than in the control group, suggesting satisfactory outcomes regarding the expression of RUNX2 in *h*PDLSCs after exposure to these sealers. This observation corroborates the findings of a previous study [40]. The expression of osteocalcin (OC) and osteoprotegerin (OPG), important markers for bone synthesis [57,58], was similar to the expression of RUNX2; however, the expression of the OPG gene was higher in the VDW.1Seal group than in the ADSeal group, indicating the superiority of VDW.1Seal as a calcium-silicate-based sealer on the osteogenic potential of the *h*PDLSCs compared to the epoxy-resin-based sealer ADSeal. However, in a recent study, Endosequence BC Sealer HiFlow showed higher expression of ALP, RUNX2, and OC than the other sealers [35]. The increased expression of osteogenic markers in cells treated with HCSBS has been previously reported [59,60,61].

Both calcium-silicate-based sealers presented alkaline pH values at all observation periods (Figure 8), which agrees with a recent report [62]. At day 1 and after 14 days, Endosequence BC Sealer HiFlow showed significantly higher pH values than VDW.1Seal. Additionally, regarding the calcium ion release, values for Endosequence BC Sealer HiFlow were significantly higher than those for VDW.1Seal at day 1 and after 14 days. The findings of the current study are in line with a study recently published that showed that the calcium ion release was higher in the EndoSequence BC Sealer group followed by the AH Plus Bioceramic and AH Plus with the lowest release [63]. Surprisingly, after 1 and 2 weeks, calcium ion release values for ADSeal were nearly identical to the values obtained for VDW.1Seal. This finding can be attributed to the presence of calcium in the composition of ADSeal.

As indicated previously in different studies, the rise of the pH values of sealers may be correlated with the deposition of mineralized tissue and play a role in the healing process. The alkaline pH of root canal sealers could inhibit the activity of osteoclasts and prevent demineralization of the tooth structure [64,65]. However, in a more recent study, it was found that when human mesenchymal stem cells were maintained in vitro in an external medium of moderate alkaline pH (7.90), osteogenic differentiation (particularly alkaline phosphatase [ALP] activity and RUNX2, ALP, and BSP gene expression) was unaffected; however, at higher pH values, osteogenic differentiation was severely inhibited [66]. At alkaline (>7.54) pH values, the development of mineralized nodules in the extracellular matrix of human mesenchymal stem cells was completely prevented. These results indicate that the osteogenic differentiation of osteoprogenitor cells is negatively affected by excessive alkalinization in the microenvironment [66]. These results may explain why GuttaFlow-2 showed better performance regarding the osteogenic potential of stem cells than Endosequence BC Sealer HiFlow, although GuttaFlow-2 displayed lower pH levels than the other sealers. Moreover, the findings of Monfoulet et al. in 2014 [66] may elucidate why VDW.1Seal caused better osteogenic differentiation of the *h*PDLSCs than Endosequence BC Sealer HiFlow, though the pH values of VDW.1Seal were lower than those of Endosequence BC Sealer HiFlow. The bioactive potential of AH Plus Bioceramics, which is identical to the VDW.1Seal sealer, has already been shown in a recent study [40].

Investigation of the biological effect of different sealers on *h*PDLSCs is crucial and mandatory to be performed on any new sealer introduced into the market to highlight the sealers with favorable performance and consequently ensure that a better outcome could be achieved. The differences in the effect of these sealers on stem cell viability and osteogenic potential should not be attributed only to the type of the setting reaction but also to the sealer composition, ion release, solubility, and pH values.

## 5. Conclusions

GuttaFlow-2 showed the highest percentage of cell viability, the significantly lowest percentage of the open wound, and the significantly highest absorbance of Alizarin Red, and both Gutt-Flow-2 and VDW.1Seal displayed the significantly highest osteogenic marker gene expression of all sealers investigated. The results obtained for GuttaFlow-2 suggest that the pH value and calcium ion release should not be regarded as the only key factors for the osteogenic differentiation of osteoprogenitor cells. Endosequence BC Sealer HiFlow showed the significantly highest calcium ion release. In general, the findings of this in vitro study imply that both GuttaFlow-2 and the newer VDW.1Seal are eligible sealers for root canal obturation.

## Figures and Tables

**Figure 1 jcm-12-02344-f001:**
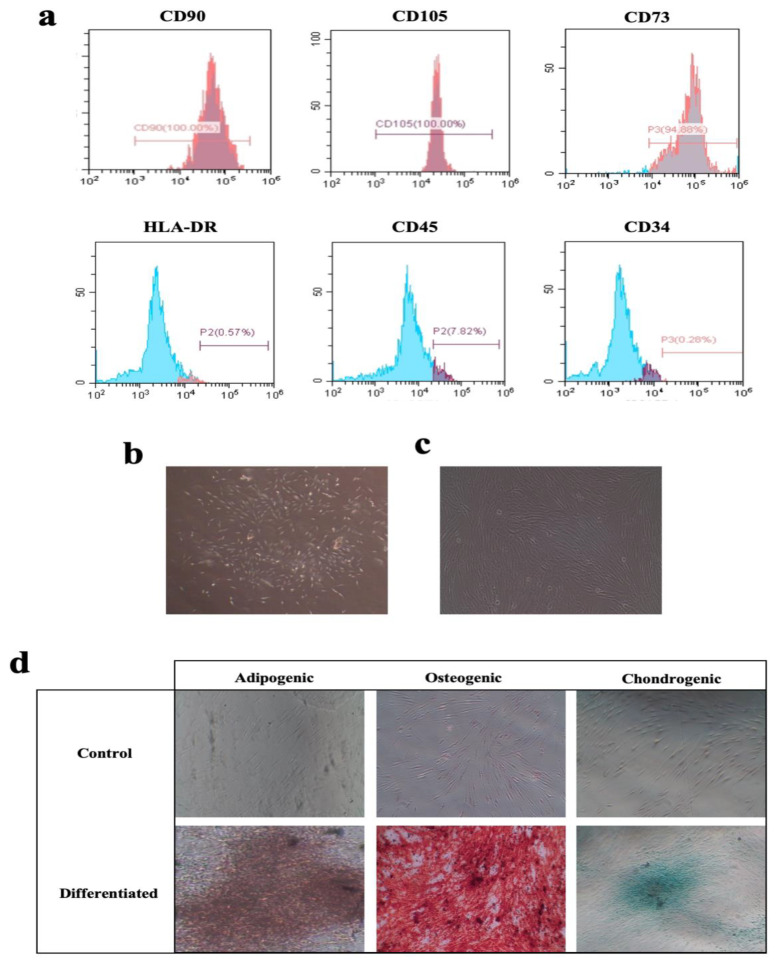
(**a**). Flowcytometry analysis showing that 100% of the hPDLSCs were CD105-positive, 100% CD90-positive, and 94.88% were CD73-positive, respectively, while only 7.82% of the hPDLSCs were CD45-positive, 0.28% were CD34-positive, and 0.57% were HLA-DR-positive. (**b**). Clone formation at around 5 days of culture (×40 original magnification). (**c**). Spindle cell morphology of hPDLSCs (×100 original magnification). (**d**). Oil Red O staining after adipogenic differentiation, Alizarin Red S (ARS) staining after osteogenic differentiation, and Alcian Blue staining after chondrogenic differentiation (×40 original magnification).

**Figure 2 jcm-12-02344-f002:**
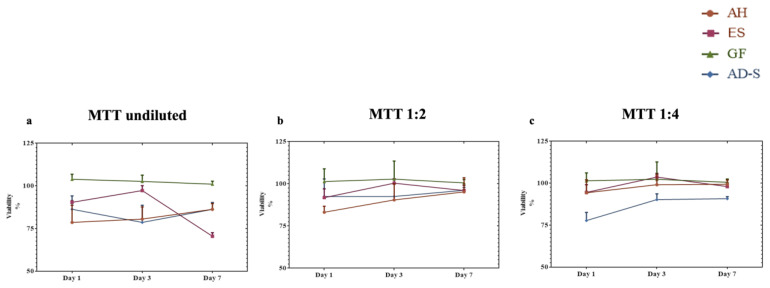
MTT assay showing the mean percentage of cell viability with standard deviation after exposure to the different sealers in medium. Data obtained by each group was normalized based on cells + medium (control). (**a**) Undiluted eluates. (**b**) Dilution 1:2. (**c**) Dilution 1:4. VDW.1—VDW.1Seal, ES—Endosequence BC Sealer HiFlow, GF—GuttaFlow-2, AD-S—ADSeal.

**Figure 3 jcm-12-02344-f003:**
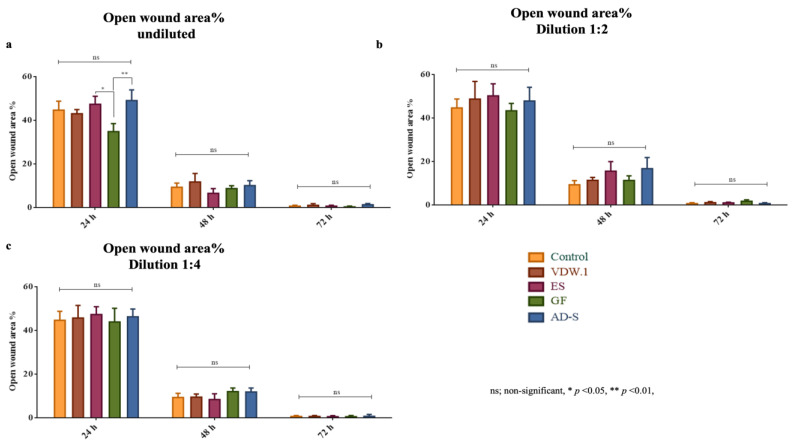
Percentage of open wounds in eluates of the different sealers used during 24 h, 48 h, and 72 h, given as means and standard deviations. (**a**) Undiluted eluates. (**b**) Dilution 1:2. (**c**) Dilution 1:4. VDW.1; VDW.1Seal, ES; Endosequence BC Sealer HiFlow, GF; GuttaFlow-2, AD-S; ADSeal.

**Figure 4 jcm-12-02344-f004:**
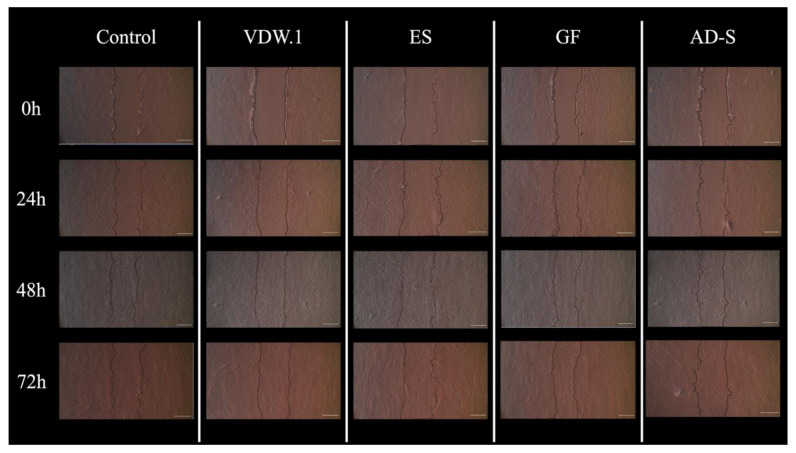
Images of the scratch healing of all groups acquired at 0, 24, 48, and 72 h with a phase-contrast microscope (TCM 400) at magnification of 40×. Cell migration area was calculated using Image J software and wound closure percentage was calculated for each well. VDW.1—VDW.1Seal, ES—Endosequence BC Sealer HiFlow, GF—GuttaFlow-2, AD-S—ADSeal.

**Figure 5 jcm-12-02344-f005:**
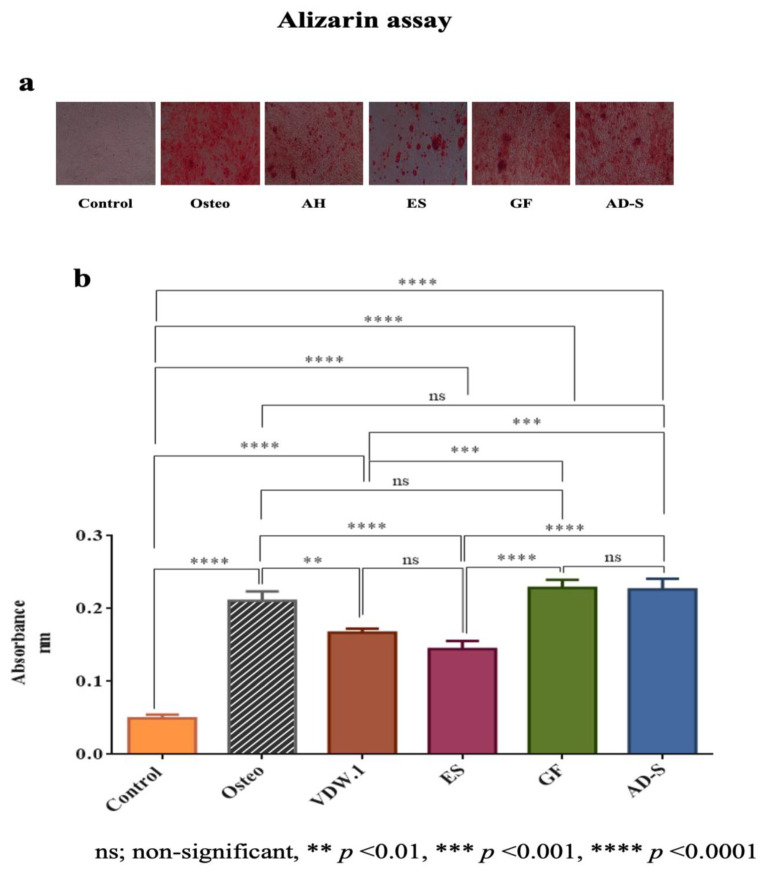
(**a**) Alizarin Red staining for the different sealers (×40 original magnification). (**b**) Bar graph of mean with standard deviation showing the absorbance of Alizarin Red S staining [nm] as an indication of calcium nodules deposition. Osteo—osteogenic differentiation media, VDW.1—VDW.1Seal, ES—Endosequence BC Sealer HiFlow, GF—GuttaFlow-2, AD-S—ADSeal.

**Figure 6 jcm-12-02344-f006:**
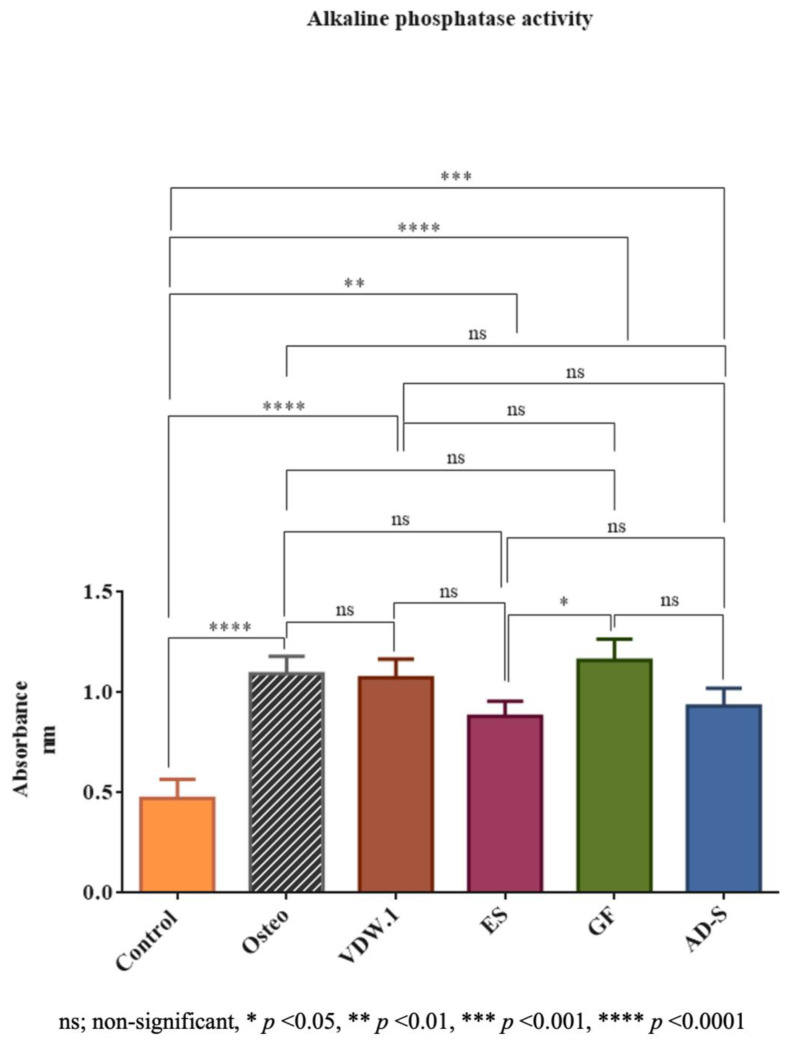
Photometric absorbance of the alkaline phosphatase activity assay. Given are the means with standard deviations. Osteo—osteogenic differentiation media, VDW.1—VDW.1Seal, ES—Endosequence BC Sealer HiFlow, GF—GuttaFlow-2, AD-S—ADSeal.

**Figure 7 jcm-12-02344-f007:**
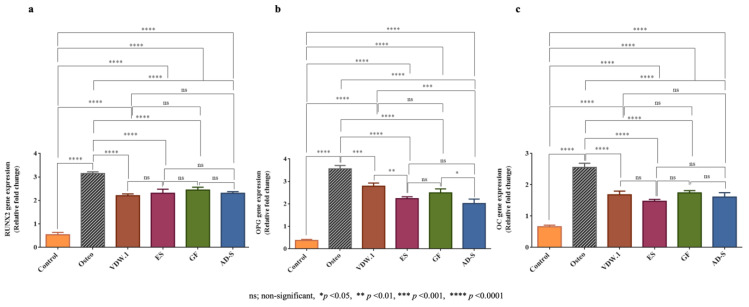
Relative gene expression for the different groups, given as means and standard deviation. (**a**) RUNX2 gene expression between the different sealers examined and the negative and positive control groups. (**b**) OC gene expression. (**c**). OPG gene expression.Osteo—osteogenic differentiation media, VDW.1—VDW.1Seal, ES—Endosequence BC Sealer HiFlow, GF—GuttaFlow-2, AD-S—ADSeal.

**Figure 8 jcm-12-02344-f008:**
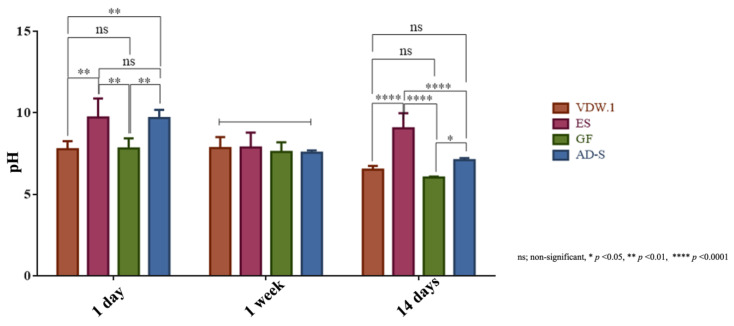
pH values of the different sealers, given as means and standard deviations at day 1, after 1 week, and after 14 days. VDW.1—VDW.1Seal, ES—Endosequence BC Sealer HiFlow, GF—GuttaFlow-2, AD-S—ADSeal.

**Figure 9 jcm-12-02344-f009:**
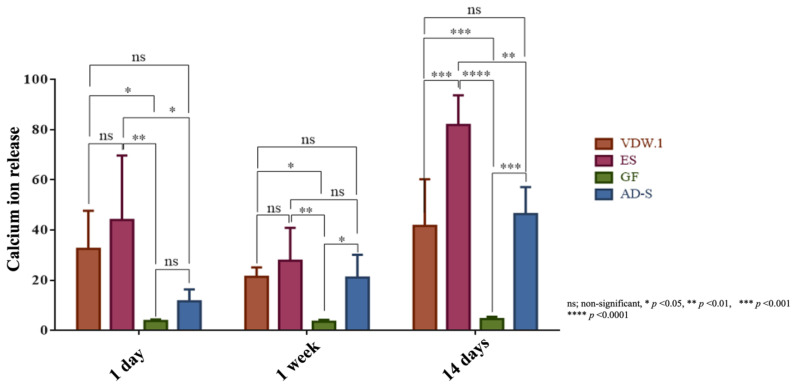
Calcium ion release values of the different sealers, given as means and standard deviations at day 1, after 1 week, and after 14 days. VDW.1—VDW.1Seal, ES—Endosequence BC Sealer HiFlow, GF—GuttaFlow-2, AD-S—ADSeal.

**Table 1 jcm-12-02344-t001:** Composition of sealers used in this study.

Sealer	Manufacturer	Composition
**VDW.1Seal**	Dentsply Sirona, Munich, Germany	Zirconium dioxide, tricalcium silicate, dimethyl sulfoxide, lithium carbonate, thickening agents
**Endosequence** **BC Sealer HiFlow**	Brasseler USA Savannah, GA, USA	Zirconia, dicalcium silicate, tricalcium silicate, calcium hydroxide, and fillers
**GuttaFlow-2**	Coltene Whaledent, Langenau, Switzerland	Gutta-percha powder, polydimethylsiloxane, silicone oil, zirconium dioxide, platinum catalyst, micro-silver, paraffin oil.
**ADSeal**	Meta Biomed, Cheongju, Korea	Base paste: Epoxy oligomer resin, ethylene glycol salicylate, zirconium oxide, calcium phosphate, bismuth subcarbonate Catalyst paste: Poly aminobenzoate, triethanolamine, zirconium oxide calcium phosphate, bismuth subcarbonate, calcium oxide

**Table 2 jcm-12-02344-t002:** The sequences of primers for the osteogenic markers used in the experiment.

	Forward Sequence	Reverse Sequence	Gene Accession Number
**RUNX2**	GTTATGAAAAACCAAGTAGCCAGGT	GTAATCTGACTCTGTCCTTGTGGAT	NM_009820
**OC**	TTCATGTGGGGTGTCTCTGA	CTGGGCCTTGGTCTTGAGT	M23637.1
**OPG**	CTAATTCAGAAAGGAAATGC	GCTGAGTGTTCTGGTGGACA	NM_012870
**β-actin**	TCCGTCGCCGGTCCACACCC	TCACCAACTGGGACGATATG	NM_031144.3

RUNX2: Runt-related transcription factor 2, OC: Osteocalcin, OPG: Osteoprotegerin.

## Data Availability

The datasets generated during and/or analyzed during the current study are available from the corresponding author on reasonable request.

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
