# Peer review of "Effect of Different Sealers on the Cytocompatibility and Osteogenic Potential of Human Periodontal Ligament Stem Cells: An In Vitro Study"

_jcm, 2023, doi:10.3390/jcm12062344_

Round 1

Reviewer 1 Report

An interesting study regarding the endodontic root canal sealers and their influence over the periapical biology. In my opinion, this manuscript can be improved with some information related to the root canal filling technical concepts, which are of importance for a practitioner.  

Pg.1, lines 35-38. For a reader less familiarized with the subject, it would be of interest a more detailed description of the Scanidinavic and North American Schidler endodontic concept regarding the root filling technique (where and why the filling material/sealer should stop). Moreover, it must be emphasized that not every time the sealer/gutta extruded in the periapical space a periapical reaction appears.

Pg.2, lines 90-91. It would be of interest for the reader a small description of the technique and whom belongs ‘’…was performed as described by [20]….’’

Pg.13-14, lines 311-312. A more detailed description of the two root canal filling technique concepts and it must be emphasized that not every time the sealer/gutta extruded in the the periapical space a periapical reaction appears (the periapical reaction is usually linked more to the individual reactivity of each organism and the lack of disinfection)  - clinically is important for a practitioner

Author Response

Reviewer #1 comments to the authors:

  1. Comment

Pg.1, lines 35-38. For a reader less familiarized with the subject, it would be of interest a more detailed description of the Scanidinavic and North American Schidler endodontic concept regarding the root filling technique (where and why the filling material/sealer should stop). Moreover, it must be emphasized that not every time the sealer/gutta extruded in the periapical space a periapical reaction appears.

Response: this part has been added in the introduction section with the reference to the study mentioned in pg.1&2 lines 35-68.

  1. Comment

Pg.2, lines 90-91. It would be of interest for the reader a small description of the technique and whom belongs was performed as described by (20]....”

Response: description of the technique and whom it belongs were added in methodology section pg.3 lines 134-140.

  1. Comment

Pg. 13-14, lines 311-312. A more detailed description of the two root canal filling technique concepts and it must be emphasized that not every time the sealer/gutta extruded in the periapical space a periapical reaction appear (the periapical reaction is usually linked more to the individual reactivity of each organism and the lack of disinfection) clinically is important for a practitioner.

Response: this part has been added in the discussion section in pg.14 lines 375-378.

Reviewer 2 Report

The authors presented the cytotoxic behavior of some endodontic sealers, as previously, the cell behavior has been detected. However, in this study, the authors compared and extensively analyzed. There are comments, which should be addressed.

How did the sample size calculation do?

What were the dimensions of the discs?

Why were dilutions made only 1:1, 1:2, and 1:4. The authors should do more dilutions e.g. 1:8, 1:16, 1:32, and 1:64, to get better reliability of the results.

Provide Ethical permission reference number.

Which teeth were extracted: molars / premolars?

The cell study was done in set mode, what about in unset mode, as the sealers interact immediately with the adjacent cells? The authors should perform a study and compare the set and unset behavior.

Author Response

Reviewer #2 comments to the authors:

  1. Comment

How did the sample size calculation do?

Response: Number of discs used (sample size) was calculated according to the material surface area / medium volume ratio, 3 mm2/ml approximately, per ISO10993-12 standards (added in methodology section pg.3 line 121).

  1. Comment

What were the dimensions of the discs?

Response: Discs dimensions are added in methodology section pg3 line 115.

  1. Comment

Why were dilutions made only 1:1, 1:2, and 1:4. The authors should do more dilutions e.g. 1:8, 1:16, 1:32, and 1:64, to get better reliability of the results.

Response:

In Cell Viability Assay (MTT Assay)

Cells viability using different materials eluates were determined in comparison to the control group (medium without eluates).

The dilutions used in this study demonstrated more than 70% cell viability (The ISO limit for defining a cytocompatible material) when compared to the control. No significant differences were found in cell viability between the (1:2) and (1:4) dilutions at day 7 and cells viability was approximately 100%. Further dilutions are expected to show similar viability. Further dilutions may be needed if there is a significant cytotoxic effect of the eluates on stem cells, which was not the case in our study. In addition, our work was guided by several studies that used 1:4 as a final dilution (1,2).

  1. Comment

Provide Ethical permission reference number.

Response: Ethical permission reference number is added pg.3 line 126.

  1. Comment

Which teeth were extracted: molars / premolars?

Response: Molars were used and added in methodology section pg.3 line 124.

  1. Comment

The cell study was done in set mode, what about in unset mode, as the sealers interact immediately with the adjacent cells? The authors should perform a study and compare the set and unset behavior.

Response: we appreciate this valuable comment, however in this study we aimed to mimic the clinical situation where the sealers are inserted inside the canals in a form that undergoes setting in short time either by mixing different pastes or by exposure to air and moisture. So, we studied the form (set form) that will remain for an extended periods in contact with the periapical tissues in case of sealer extrusion while the unset form might be washed out by tissue fluids in short time. As mentioned in the discussion (page 14 line 404-409), Future studies should include both preparations to provide a comprehensive biological description of the examined materials.